# Deposition of Thin Electroconductive Layers of Tin (II) Sulfide on the Copper Surface Using the Hydrometallurgical Method: Electrical and Optical Studies

**DOI:** 10.3390/ma16145019

**Published:** 2023-07-15

**Authors:** Anna Komenda, Marek Wojnicki, Dzmitry Kharytonau, Grzegorz Mordarski, Edit Csapó, Robert P. Socha

**Affiliations:** 1Faculty of Non-Ferrous Metals, AGH University of Science and Technology, Mickiewicza Ave. 30, 30-059 Krakow, Poland; 2CBRTP SA Research and Development Center of Technology for Industry, Ludwika Waryńskiego 3A, 00-645 Warszawa, Poland; dmitry.kharitonov@cbrtp.pl (D.K.);; 3Soft Matter Nanostructures Group, Jerzy Haber Institute of Catalysis and Surface Chemistry, Polish Academy of Sciences, Niezapominajek 8, 30-239 Krakow, Poland; 4MTA-SZTE “Lendület” Momentum Noble Metal Nanostructures Research Group, University of Szeged, Rerrich B. Sqr. 1, H-6720 Szeged, Hungary

**Keywords:** thin films, synthesis, conductivity, solar panel, photovoltaics

## Abstract

Thin films of tin (II) sulfide (SnS) were deposited onto a 500 µm thick copper substrate by a chemical bath method. The effect of sodium (Na) doping in these films was studied. The synthesis of the films was performed at temperatures of 60, 70, and 80 °C for 5 min. The microstructure of the SnS films analyzed by scanning electron microscopy (SEM) showed a compact morphology of the films deposited at 80 °C. The edges of the SnS grains were rounded off with the addition of a commercial surfactant. The thickness of different SnS layers deposited on the copper substrate was found to be 230 nm from spectroscopic ellipsometry and cross-section analysis using SEM. The deposition parameters such as temperature, surfactant addition, and sodium doping time did not affect the thickness of the layers. From the X-ray diffraction (XRD) analysis, the size of the SnS crystallites was found to be around 44 nm. Depending on the process conditions, Na doping affects the size of the crystallites in different ways. A study of the conductivity of SnS films provides a specific conductivity value of 0.3 S. The energy dispersive analysis of X-rays (EDAX) equipped with the SEM revealed the Sn:S stoichiometry of the film to be 1:1, which was confirmed by the X-ray photoelectron spectroscopy (XPS) analysis. The determined band-gap of SnS is equal to 1.27 eV and is in good agreement with the literature data.

## 1. Introduction

Thin-film solar cells offer distinct advantages over crystalline silicon (Si) cells, primarily due to the wide range of materials available for constructing multi-junction systems [1]. These systems have demonstrated enhanced efficiencies comparable to Si modules [2,3,4,5]. Furthermore, thin-film absorbers with a direct band gap exhibit higher optical absorbance per unit of the material compared to monocrystalline Si [6,7]. Importantly, the production of thin-film solar cells requires a smaller budget, resulting in lower energy consumption during the manufacturing process.

In contrast to conventional Si solar cells, thin films can be applied to flexible substrates [8,9,10], such as polyethylene terephthalate (PET), significantly reducing the overall weight of the solar cell. This property is particularly valuable for building integrated photovoltaic systems.

Tin (II) sulfide (SnS) has emerged as an attractive absorption material for low-cost thin-film solar cells due to its desirable properties, including a band gap of 1.3 eV, high carrier mobility, high light absorption coefficient, and good p-type conductivity [11,12]. According to Prince–Loferski diagrams, devices based on SnS mono junctions have a theoretical conversion efficiency of 24%. Additionally, SnS compounds are non-polar, reducing the number of unsaturated bonds on the surface and lowering reactivity [13].

The electrical and optical properties of SnS can be modified by optimizing growth conditions or by doping the compound with appropriate dopants in the SnS crystallographic network [14]. Metal doping, for instance, alters the energy gap and band positions in the SnS band structure, thereby determining the electrical properties of the films through metal substitution in the Sn vacancies, which increases carrier concentration to the range of 10^15^–10^18^ cm^−3^ [15].

In the scientific literature, various methods of SnS deposition have been reported, including spray pyrolysis, electrodeposition, reaction sputtering, chemical vapor deposition, radiofrequency sputtering, vacuum vapor deposition, and sulfidation of pre-deposited tin precursor layers. Although these methods are scalable, they possess certain disadvantages when it comes to the SnS deposition. For instance, vacuum vapor deposition allows precise control of process parameters and the deposition of high-purity deposits, but it requires special crucibles, high temperatures, and a controllable atmosphere, which limits its industrial implementation. Pyrolysis, commonly used in industry, necessitates a minimum temperature of 350 °C for the deposition of tin sulfide compounds. Chemical vapor deposition, another widely used technique, requires an expensive setup and high temperatures, as well as careful consideration of precursor gas purity. Furthermore, depositing layers over larger areas can be challenging.

It is important to note that achieving a 1:1 ratio of elements in SnS using thermal methods is challenging due to the high volatility of sulfur. Sulfur deficiency can result in the migration of tin to intergrain boundaries, interfaces, or interstitial positions, leading to incomplete occupancy of sulfur positions. Wet or hydrometallurgical methods provide an alternative solution by allowing synthesis at lower temperatures (below 100 °C), reducing energy consumption and final production costs. These methods are relatively simple to implement in production lines, offering the possibility to modify various parameters. For instance, reducing particle size can enhance carrier mobility, absorption, photo- and thermos-stability. Another interesting observation is that SnS and SnSe crystals exhibit an orthorhombic structure (Pnma) at room temperature, transitioning to a cubic structure (Cmcm) at higher temperatures.

Pure SnS compounds typically exhibit a lamellar morphology with volume defects and poor interface characteristics. These volume defects adversely affect the open circuit voltage (OCV) and short circuit current density (JSC). Extensive studies have demonstrated the positive effects of Na doping in compounds such as n-Si/Cu_2_SnS_3_, Cs_2_AgBiBr_6_, and Cu(In,Ga)Se_2_ [16].

Furthermore, there is a dearth of information in the literature regarding Na doping of SnS films and the impact of concomitant surfactant properties on the conductivity and surface morphology of such cells deposited on a metallic substrate like copper. While there are several reports on the chemical bath synthesis of SnS using similar chemical reagents [17,18,19,20,21,22,23], detailed investigations on the influence of temperature, composition, and surfactants on the conductivity of the SnS layer are lacking.

In this work, we focus on depositing tin (II) sulfide thin films on a copper substrate. The selection of copper as a substrate is justified by its lightweight, lower cost compared to silicon, availability, and high conductivity, making it suitable as an electrical charge collector [24,25].

Hence, the aim of this work is to investigate the influence of selected parameters, such as temperature, time, and surfactant content, on the conductivity, thickness, and composition of the deposited thin SnS films. These systematic studies bring us closer to the subsequent step of optimizing the optical properties and constructing a complete photovoltaic (PV) cell.

## 2. Materials and Methods

### 2.1. Stock Solutions

Thin tin (II) sulfite films were deposited on a 500 µm-thick copper sheet (purity 99.999). Chemical reagents, such as tin (II) chloride dihydrate, thioacetamide, tartaric acid, hydrochloric acid, and deionized water were used for the chemical synthesis. A commercially available mixture of anionic surfactants, non-ionic surfactants, and amphoteric surfactants was used as the surfactant (Ludwik, Poland). For the chemical bath, 0.1 mol/dm^3^ tin (II) chloride solutions were prepared in 0.1 mol/dm^3^ hydrochloric acid (Avantor, Poland). The concentrations of the thioacetamide and tartaric acid solutions were 0.6 and 1.0 mol/dm^3^, respectively.

### 2.2. Preparation of the Copper Surface

A 30 × 5 cm^2^ sheet of copper was subjected to electrochemical polishing in a concentrated phosphoric acid (V) environment. A current density of 90 mA/cm^2^ was used. After polishing, the sample was rinsed profusely with deionized water and then with isopropanol.

### 2.3. Effect of Temperature

Initially, a temperature- controlled synthesis was performed at 60, 70, or 80 °C. For this, a prepared copper sheet was placed in the beaker so that it adhered around the wall of the beaker. A solution composed of 160 mL of stannous chloride, 80 mL of tartaric acid, and 280 mL of deionized water was then heated in the beaker until the respective temperatures were achieved. A mechanical stirrer was used for stirring (110 rpm). When the solution reached the desired temperature, 40 mL of thioacetamide was added into the beaker and the formed solution waited for another 5 min. Finally, the copper sheet was removed from the solution rinsed generously with deionized water and dried with a flow of compressed air.

### 2.4. Effect of Surfactant Addition

The syntheses were conducted at a constant temperature of 80 °C, depending on the amount of the surfactant additive. A series of experiments was performed in the solutions additionally containing 0.0038, 0.0075, 0.0150, 0.0300, 0.0600, 0.1200, or 0.2400 g of the surfactant additive. As a result, the surfactant concentrations in the chemical bath solution were, respectively: 6.78610^−3^ 1.33910^−2^; 2.67910^−2^; 5.35710^−2^; 1.07110^−1^; 2.14310^−1^; and 4.28610^−1^ g/dm^3^. The synthesis was performed analogously to that described in Section 2.3 (temperature dependence), but a relevant amount of surfactant was additionally added to the system on the stage of the thioacetamide.

### 2.5. Sodium Doping

As a next process variable, a series of sodium-doped tin(II) sulfides was made. For this purpose, sodium chloride solution (0.1 mol/dm^3^) was prepared. The syntheses were performed at 85 °C and with the addition of four droplets of the surfactant. The synthesis procedure remained the same. In the beginning, when the thioacetamide and surfactant were added and 20 mL of sodium chloride was also added, and after five min, the synthesis was completed. In another series of experiments the sodium chloride solution was successively added into the reaction mixture after 1, 2, 3, and 4 min, from the addition of the thioacetamide and the surfactant. However, the total synthesis time starting from the addition of the thioacetamide and the surfactant was 5 min. In another variant, the synthesis time, measured by adding the surfactant, thioacetamide, and sodium chloride at once, was 3 min. In the last variant, 1 min and 45 s after adding thioacetamide and surfactant, stannous chloride was added, and the synthesis was completed after another 1 min and 45 s.

### 2.6. Instruments Used for Analysis

The CasaXPS 2.3.12 software was applied for the analysis of the XPS spectra. No charging was observed for the studied SnS; therefore, no additional calibration of the spectra energy scale was applied. In the spectra, the background was approximated by a Shirley profile. The spectra deconvolution into a minimum number of components was performed by the application of the Voigt-type line shapes (70:30 Gaussian/Lorentzian product).

The analytic depth of the XPS method was estimated as 10.2 nm. The calculations were performed with QUASES-IMFP-TPP2M Ver 2.2 software according to the algorithm proposed by Tanuma et al. [26] This estimation takes into account 95% of photoelectrons escaping from the surface. The experimental error of the XPS analysis is approximately ±3%.

Top-surface and cross section scanning electron microscopy (SEM) images were taken to investigate the surface morphology and the thickness of the deposited SnS layers, respectively. The SEM images were taken using a JEOL-6000 Plus instrument (Tokyo, Japan).

The obtained coatings underwent phase analysis with the use of a Rigaku Mini Flex X-ray diffractometer applying filtered radiation of the CuKα lamp.

The thickness of the SnS layer was measured using SEM microscopy. For this purpose, the cut Cu sheets coated with the SnS layer were placed vertically in the mold, pre-pressing them into plasticine. Vertically protruding elements were then flooded with two-carbon epoxy resin (Dragon Poland, Skawina, Poland). After curing the resin, the plasticine was removed, and the excess resin was sanded off on both sides. One side was used to make electrical contact, while the other side was polished in stages. Polishing pastes of 5 and 1 µm, then 100 and 5 nm, were used. Polishing allowed to get on one side a mirror flash on the resin surface and Cu/SnS cross-sections.

The second method of estimating the thickness of the SnS layer on the Cu surface consisted of measuring the optical properties of the layer. For this purpose, a laser ellipsometer SE 400adv PV was used.

During the SnS conductivity measurements, a copper adhesive tape with a width of 10 mm and a Cu thickness of 25 µm was used. This tape protected the SnS layer against mechanical damage by the measuring court blades. The resistance of this tape is 1 Ω and is about 100 times lower than the resistance of SnS. The electrical resistance of the contacts was neglected in the calculations. A Keithley 2100/230-240 CAT II 600 V bench multimeter was used for conductivity measurements.

The energy gap of the obtained SnS layers was determined on the basis of the reflection spectrum recorded using the PerkinElmer Lambda 950 spectrophotometer equipped with a 150 mm integrating sphere.

## 3. Results

On the copper surface, tin sulfide (II) was deposited. All experimental conditions, as well as samples’ sizes, are gathered in Table 1.

### 3.1. Influence of Deposition Temperature

Figure 1a,b shows the appearance of the SnS layers deposited on the copper surface depending on the process temperature.

The layer obtained at 60 °C contains significant voids (black spots in image Figure 1a). An increase in temperature reduces the number of surface defects. In the case of the samples synthesized at 70 °C, the number of surface defects was reduced in comparison to the sample obtained at 60 °C. However, they were still present. Further increasing in the process temperature reduced the number of surface defects, as can be seen in Figure 1b.

The effect of temperature on reducing the number of defects can be associated with a change in the solution viscosity. The process is conducted in a heterogeneous system. Therefore, the rate-limiting element may be transported at the interface. In such a situation, if the rate-limiting step is diffusion, the viscosity of the solution can be changed by adding, e.g., alcohol, increasing the temperature, or increasing the intensity of mixing. However, we would like to point out that the detailed study of the mechanism is not the subject of this work, but the preliminary selection of the conditions for obtaining such layers.

The surface is homogenous, and well-defined crystal grains are visible. The SnS_PshL_2_70 sample has morphology features between two studied extreme temperatures, which is unsatisfactory and further tests at 70 °C were abandoned. The SnS layers are characterized by good adhesion to the substrate because the coating did not shell off when cutting the sample with scissors (to prepare a section for the SEM examinations). At high magnification, it was observed that the length of a single SnS crystal obtained under these conditions is about 500 nm.

### 3.2. Influence of Surfactant Addition

The results of the SEM analysis of the samples obtained in the conditions when the surfactant was added SEM in the reaction bath are shown in Figure 2. At low magnifications, it is seen that SnS layers are built up in two different ways (see Figure 2a). In particular, Figure 2a,b shows packed areas and loose layers growing as bars. This difference is explained by the preferential crystallographic direction of the SnS growth in the orthorhombic unit cell SnS grown. In turn, this is probably greatly influenced by the texture of the copper substrate.

It is worth noting that the copper sheet used for the tests was rolled. This means that the copper grains are deformed, distorted, and stretched toward the rolling direction. So, there are stresses in the base material, and this energy influences the growth of the SnS layers, which in some areas proceeds along the short side forming very dense layers, and in other areas-along the long side where the layer grows in the form of bars.

Figure 3, starting from panel a and ending with panel c, shows less and less visible differences on the surface of the sample after changing the initial surfactant concentration. Figure 3c, which shows a sample obtained using the highest surfactant concentration, shows practically no leaks in the layer, and the boundaries between the individual areas are not so visible.

In turn, Figure 3a shows the morphology of the SnS layer when the smallest concentration of surfactant is used. The layer is very uneven. It seems that it is, therefore, worth using higher concentrations of the surfactant.

Higher magnification reveals how the shape of the crystallites changes under the influence of increasing the surfactant concentration. The same samples as before are now shown at a magnification of 25,000 (Figure 4).

The lowest surfactant addition does not affect the shape of the crystallites, which are cuboidal in shape. The concentration of the surfactant at the level of 5.357 × 10^−2^ g/dm^3^ (Figure 4b) causes a slight deformation of the edges. They become slightly rounded. However, very high concentrations of the said agent cause large deformations of the crystallites, as shown in Figure 4c). Using the highest concentration of surfactant used in work (4.286 × 10^−1^ g/dm^3^) makes the crystallites no longer resemble cuboids. The surface morphology at higher magnifications resembles rose petals.

It seems that if the concentration of surfactant is increased, the circles (crystals) become closer together (integrate). Increasing the surfactant concentration may change the crystal growth direction. Such an effect is frequently observed and reported in the literature [27]. The best example is gold nanorods formation in the presence of CTAB (Cetyltrimethylammonium bromide).

### 3.3. Influence of Sodium Addition

In this work, in further research (doping with sodium), it was decided to use a higher addition of surfactant, causing the growth of crystallites in the shape of rose petals. The appearance of the surface of the selected sample, in which the surfactant concentration of 2.143 × 10^−1^ g/dm^3^ was applied, is shown below (Figure 5).

The decision to use such an amount of the additive was made because it guaranteed a thorough, tight coverage of the copper substrate (Figure 5a). An interesting observation is comparing the appearance of areas with different types of coverage, i.e., different, dominant crystallographic directions and layer growth. The appearance of these areas is shown in Figure 5b,c. In addition, Figure 5c shows that the thickness of such a single flake, under these synthesis conditions is 75 ± 20 nm.

### 3.4. Results of EDAX and SEM Analyses for Sample Cross-Sections

To determine the SnS layers’ thickness on copper, the samples’ cross-sections were prepared. Unfortunately, the boundary between Cu and SnS is not clearly visible. Therefore, line scans were performed on the cross-section of the sample to identify this boundary. Figure 6 shows the EDAX analysis of SnS_Na_B011 sample.

Based on SEM images, layer thickness measurements were made, taking into account the results of the EDAX analysis. To test the repeatability of the layer thickness measurement, 50 measurements were made in the Image J^®^ program for one selected sample (SnS_Na_B010). The graph of the frequency function of this distribution is shown in Figure 7.

Table 2 contains the results of measurements of the thickness of SnS layers on the copper substrate derived from the analysis of SEM images using the Image J^®^ program.

Increasing the temperature of the chemical bath to 80 °C resulted in a decrease in the thickness of the SnS layer. This is an expected observation because with increasing temperature, the nucleation rate increases, which begins to exceed the rate of the propagation stage significantly. This means a shortened crystallite growth time in these conditions. It is worth noting, however, that the standard deviation (S.D.) has also changed for the analyzed results. Based on the average values and a simplified statistical analysis, it can be concluded that temperature has no statistically significant effect on the thickness of the formed SnS layer.

The thickness of the layers increased with the increase in the surfactant content. This could be since surfactants, consisting of a hydrophilic “head” and a hydrophobic “tail”, increase the stability of the particles and the growing layer.

The concentration of sodium ions in the solution during the doping was constant, so only the time of presence of sodium ions in the chemical bath was manipulated at a constant synthesis time. However, there is no significant effect of sodium doping time on the thickness of the SnS layers.

### 3.5. Results of Ellipsometric Analysis—Determination of SnS Layer Thickness

The results summarized in Table 3 represent the average layer thickness over an area much larger than the SEM analysis. The considered area in SEM images was about 3 µm^2^, and in ellipsometry, the diameter of the spot is about 200 µm^2^. Therefore, in some cases ellipsometry data is more reliable and valid for industrial applications of such thin layers. The results are consistent within the uncertainty limits of the SEM method and ellipsometry.

Based on the measurement results, it can be concluded that the manipulation of temperature, surfactant concentration, or sodium doping time does not significantly affect the thickness of the tin (II) sulfide layer, and the average thickness for all samples is around 230 nm.

### 3.6. XRD Analysis of the SnS Deposits

XRD patterns of the samples at different temperatures of the deposition process are shown in Figure 8. In the upper right corner of each diffraction pattern, there is an approximate section of the graph with reflexes identified as tin (II) sulfide.

The preferred direction of crystal growth strongly depends on the crystallographic configuration of the substrate. The substrate was rolled copper. Its crystallographic structure is deformed and irregular. The copper substrate was used in crystalline form as received. No recrystallization process was used.

The noise-to-peak ratio decreases with temperature, and the intensity of the tin (II) sulfide reflex decreases. Based on this observation, it can be concluded that the degree of crystallinity of tin (II) sulfides on the copper substrate decreases with increasing temperature.

In the above-described cross-sectional studies using the scanning microscope, it was concluded that the size of the crystallites decreases with increasing temperature. On the other hand, as the size of the crystallites decreases, the degree of crystallinity decreases. X-ray diffraction analysis allows us to study only crystalline bodies. Therefore, the disappearance of the reflection coming from tin (II) sulfide is observed.

Below, Figure 9 shows diffractograms of tin (II) sulfides depending on the surfactant concentration in the chemical bath at 80 °C.

It is observed that the addition of the surfactant has a positive effect on the degree of SnS crystallinity, despite the use of the highest examined temperature of the chemical bath used in the tests. The stabilization of the SnS molecules can explain the improvement in the degree of crystallinity. However, this positive effect disappears when the surfactant concentration is too high, as can be seen in the diffraction pattern of the sample obtained in the chemical bath with the highest surfactant concentration. The results of the XRD and SEM analyses complement each other, confirming the decrease in the degree of crystallinity at high surfactant contents in the chemical bath.

Below (see Figure 10), diffractograms of samples were collected, based on which the influence of sodium doping was examined.

### 3.7. Conductivity Measurement of SnS

It was impossible to measure the resistance of all samples because many of them had too high a resistance. The results for the samples with measurable conductivity are summarized in Table 4.

The specific conductivities were very low, except for the sample SnS_PshL_103, for which the specific conductivity was determined at about 147 S/m. This value is very different from the other values, so it is assumed that abnormalities, such as leakage of the SnS layer, have occurred. Typically, to avoid mechanical damage to the SnS layer, the SnS layer was covered with a 1 × 1 cm^2^ conductive copper tape. The measurement was performed by applying the measuring tip to the Cu tape surface. Experimentally, it was found that this brings an error of less than 1%. Without Cu tape, 90% of the measurements resulted in SnS breakdown. In addition, the measures depended on the pressure of the measuring tip on the SnS surface, which suggests that this layer was subject to mechanical deformation.

Based on the conductivity measurements, which are directly related to the band-gap, it can be concluded that Na doping is very important. However, there is no clear correlation between doping time and conductivity.

### 3.8. XPS and EDAX Analysis of Selected Samples

For one exemplary sample, EDAX and XPS analyses were performed. The EDAX measurements (Figure 11) suggest that stoichiometric tin(II) sulfide was obtained.

However, it is worth noting that the EDAX results cannot be treated as direct evidence, mainly due to the low sensitivity of this technique. For this reason, highly specialized and sensitive XPS analyses were performed for the selected sample (Figure 12).

XPS analysis allows us to determine the type of bonds and composition of the surface layer with an analytical depth of approximately 7 nm. Therefore, a part of carbon (28.8 at.%) and oxygen (27.5 at.%), some tin amount (26.6 at.%), and sulfur (17.1 at.%) is found. The high-resolution spectra (Figure 12) confirm that sulfur is present only in the form of sulfides [28]. This is supported by the value of electron binding energy (BE) at a maximum of S 2p_3/2_ core excitation at 161.6 eV (Figure 12c). On the other hand, the Sn 3d spectrum (Figure 12b) reveals three doublet components. The A component (5.1%) of Sn 3d_5/2_ excitation at 484.4 eV is assigned to metallic tin bonding to the copper substrate surface. The most intensive B component (74.6%) at BE of 486.3 eV is ascribed to tin(II) sulfide. Such assignation to SnS is supported by the value of maximum Sn M_4_N_45_N_45_ Auger excitation (Figure 12a) at the kinetic energy of 434.9 eV, which gives modified Auger parameter α’ equal to 921.2 eV, where α’ = BE (Sn 3d_5/2_) + KE (M_4_N_45_N_45_). The Auger parameter value clearly shows that the main compound at the surface is SnS. The spectrum analysis suggests also that the surface of SnS or the copper/tin interface is slightly oxidized (C component of Sn 3d spectrum (Figure 12b), 20.3% intensity, BE at 487.6 eV).

### 3.9. Band Gap Energy Determination Based on UV-Vis Spectra

The band-gaps energy of semiconductors describes the energy needed to excite an electron from the valence band to the conduction band. An accurate determination of the band gap energy is crucial in predicting the photophysical properties of semiconductors. Therefore, we performed the analysis of the band gap in the obtained SnS. For this purpose, we used the methodology described by W. Macyk et al. [29] The calculations assume that the SnS has a direct band gap. The result obtained is shown in Figure 13.

The determined band gap is 1.27 eV. The determined value is consistent with the literature data [30].

## 4. Discussion of Results

As indicated by Gisa Grace Ninan et al. [31], who obtained thin SnS layers doped with copper by spray pyrolysis, the reflex in the diffraction pattern at the 2θ position of about 31.5° comes from SnS growing in the (111) direction.

Electrodeposition of SnS layers was undertaken by Dhanasekaran Vikraman and co-authors [32]. The size of the SnS crystallites obtained by this method gives results comparable to those obtained in this work because they measured 45 nm. It is worth mentioning that the authors proved that the grain size increased with the increase in pH and that the addition of EDTA significantly reduced the size of the crystallites to even 25.3 nm. In another work using this method, Jim et al. [33] studied the effect of the temperature of the SnS growth process on the sulfur-to-tin ratio. They confirm that a ratio close to 1:1 was achievable above 70 °C. This observation is consistent with the results of studies on obtaining SnS from the solution, as the XPS and EDAX methods confirm the Sn:S ratio of 1:1 at 80 °C.

Using reactive sputtering to synthesize SnS, Lianbo Zhao and co-authors [34] obtained an SnS layer with almost no impurities of other sulfur and tin compounds only at 400 °C. At room temperature, they obtained SnS with small grains and low crystallinity. The increase in the process temperature to 400 °C caused the grains to have greater crystallinity and size. They had the highest light absorption coefficient in the wavelength range from 400 to 800 nm and the SnS layer thickness of 260 nm allowed for the absorption of 90% of incident photons. Hence, the conclusion is that the crystallinity of the SnS layers obtained in this paper should be improved so that they absorb light better. Comparing the above results with those obtained in the work, one can notice a much higher specific conductivity of SnS layers obtained using the “wet” method and almost four times thinner than SnS layers.

Dar, Govindarajan, and Dar [35], confirmed that the growth of SnS crystals in the (111) direction during the solvothermal synthesis corresponds to the increase in the reflex intensity at 2θ position of 31.49° in XRD patterns. In the vicinity of this 2θ position, there are also the most intense reflections in the diffraction patterns obtained in this work. The size of the crystallites obtained by the authors of the work [35] was 28 nm for pure SnS, and doping with chromium caused the growth of crystallites. SEM images prove that SnS grains in solvothermal synthesis have a spherical shape, which is significantly different from those obtained in the chemical bath. Atomically, the Sn:S ratio was approximately 0.8:1, so the presented work obtained a more stoichiometric SnS compound.

E. Guneri et al. [36], using a chemical bath to deposit thin SnS layers on glass, also indicated that the reflex at the 2θ position of 31.601° accounts for the SnS in the (111) direction. They used tin (II) chloride dihydrate, ammonium chloride, triethyleneamines, thioacetamide, and sodium citrate for the chemical bath. SnS grew in the form of spherical needle-shaped puffs. The conductivity of the SnS layers in the best case (synthesis time equal to 6 h) was 4.6 × 10^−3^ S/m, and the size of the crystallites in the (110) direction was 24 nm. Thus, the conductivity was as much as two orders of magnitude lower than in the presented work. The authors failed to obtain an Sn/S ratio of 1 (approx. 1.49).

A less complicated composition of the chemical bath for the deposition of SnS layers on glass was used by Sreedevi Gedi and his co-workers [21]. They used only tin(II) chloride dihydrate, thioacetamide, and tartaric acid. This work used hydrochloric acid to control the pH of the solution and prevent tin(II) chloride solution from hydrolyzing. Researchers studying the effect of temperature from 40 °C to 80 °C obtained the best optical and electrical properties at 70 °C, for which the specific conductivity was about 2.2 × 10^−4^ S/m. Compared to the presented results, this gives a worse outcome by as much as three orders of magnitude.

Chao Gao and Honglie Shen [37], who applied tin (II) chloride, ammonium citrate, sodium thiosulfate, and a few drops of ammonia water to the chemical bath, also report that the reflex at the 2θ position equal to 31.6° in the diffraction pattern is responsible for the SnS towards (111). Table 5 shows the process parameters and characteristics of the SnS thin films deposited on the glass obtained by the researchers.

The average size of the crystallites was 25–30 nm, about 15–20 nm smaller than those synthesized in this work. The authors of the work also obtained the Sn:S ratio close to 1:1 at 80 °C. As in other works, where tin (II) sulfide was deposited on glass, conductivity was much lower than in the case of the layers synthesized on a copper substrate.

The previously mentioned Mutsumi Sugiyama, Tsubasa Yokoi, and co-workers [38], who described the effect of sodium doping of thin SnS layers in the sulfidation process, indicated that the average crystallite size was 27 nm. Crystallites without admixture turned out to be smaller, which is also confirmed by this research. The thickness of the sodium-doped SnS layer was about 600 nm in the highest quality cell, which is more than twice as thick as those obtained in the work. The cross-section of the sample after 30 min of sulfidation is quite similar to the cross-section of samples.

S. Sebastian [39], who, as mentioned earlier, obtained thin SnS:Pb layers deposited on glass substrates at 350 °C by the NSP method, obtained SnS layers in which the main growth direction was (111). The size of the crystallites ranged from 13 to 41 nm. The specific conductivity was about 0.33 S/m, comparable to the obtained results.

Considering the above, it can be concluded that the presented results show that it is possible to quickly and cheaply produce the SnS layer on Cu layers using the wet (hydrometallurgical) method. The obtained layers are characterized by good homogeneity and high conductivity. None of the tested parameters significantly affected the thickness of the layer but adding a surfactant significantly improved its uniformity. This is especially important when scaling processes. Further work will be performed to create a full solar cell using the presented results.

## 5. Conclusions

In the tested range, the effect of temperature is negligible on the size of the crystallites, but the effect on the quality of the SnS layer, produced on a copper sheet using the “wet” method, is visible. The strong effect of the process temperature was noted. At 80 °C, tighter SnS layers are obtained than at lower temperatures, which was considered. However, no significant effect on the thickness of the layers is observed. Ellipsometer measurements indicated that the thickness of the SnS layer at 80 °C was approx. 222 nm, and at 70 °C it was approx. 227 nm. XRD studies additionally indicate that with the increase in the temperature of the synthesis process of SnS layers on the copper substrate, the intensity of the reflection coming from SnS decreases, indicating a decrease in the degree of crystallinity. The effect of temperature on reducing the number of defects should be associated with a change in solution viscosity. The process is conducted in a heterogeneous system.

The effect of commercial detergent on the surface morphology was also analyzed. A clear dependence of the shape of the grains on the concentration of the surfactant additive in the solution was observed. As the concentration of the commercial detergent increased, the shape of the grains changed from cuboidal to rose petal-like grains. The surfactant causes the edges of the grains to round off. The surfactant made the layers tighter, and it was visually observed that the SnS layers had a higher gloss than those from the chemical bath without the surfactant. The ellipsometry does not indicate any significant changes in the thickness of the layer. The XRD analysis did not show a clear effect of this factor on the change in the degree of crystallinity of the SnS layers, or the size of the crystallites, which are comparable to the size of the crystallites in the SnS layers that were synthesized without the addition of a commercial detergent. It seems that if the concentration of surfactant is increased, the circles (crystals) become closer together (integrate). Increasing the surfactant concentration may change the crystal growth direction. Such an effect is frequently observed and reported in the literature.

Doping with sodium affects the change in the size of the crystallites and the degree of crystallinity. The shape of the grains, however, is unchanged. The addition of sodium can reduce the size of the crystallites from 44.4 nm to 37.7 nm if the thioacetamide and doping agent are added at the same time. If the doping agent is added later, a difference of 1 min makes the SnS X-ray reflex almost imperceptible, and it is impossible to fit this reflex with the Gaussian equation to calculate the size of the crystallites using the Scherrer equation. Ellipsometry also this time showed no clear differences between samples from other series (depending on the temperature or the amount of surfactant).

Conductivity measurements do not show any dependence on any of the factors considered, and there are no significant differences in conductivity between sample series. The conductivity of the samples oscillates around 0.3 S/m.

## Figures and Tables

**Figure 1 materials-16-05019-f001:**
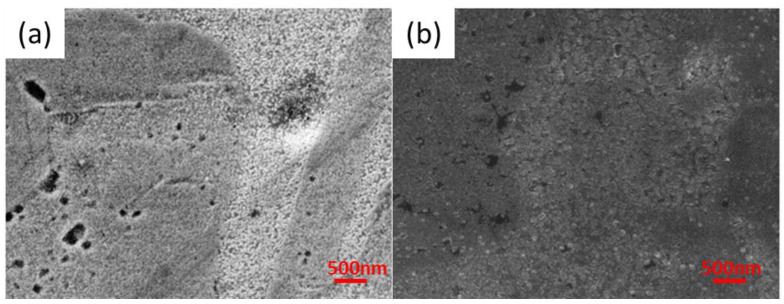
Surface morphology of SnS_PshL_2_60 (**a**) and SnS_PshL_80 (**b**) layers.

**Figure 2 materials-16-05019-f002:**
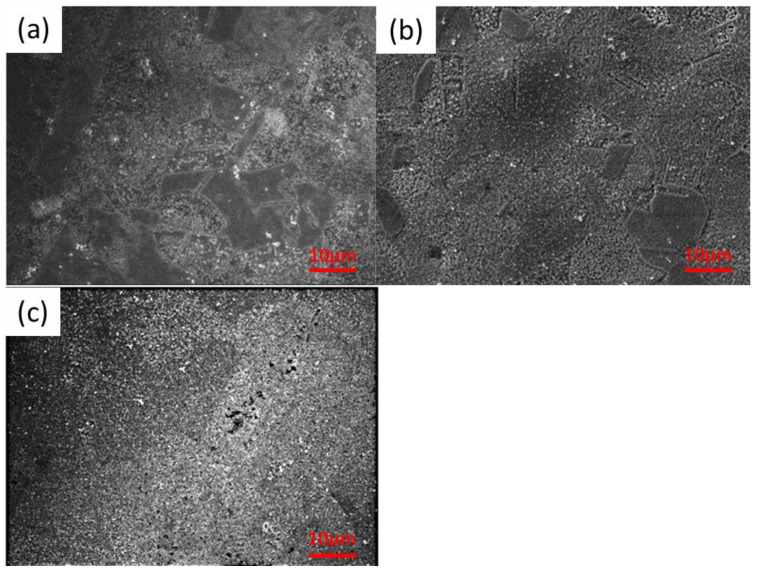
Surface morphology of the following samples: (**a**) SnS_PshL_104, (**b**) SnS_PshL_101, and (**c**) SnS_PshL_114 at magnification 1500×.

**Figure 3 materials-16-05019-f003:**
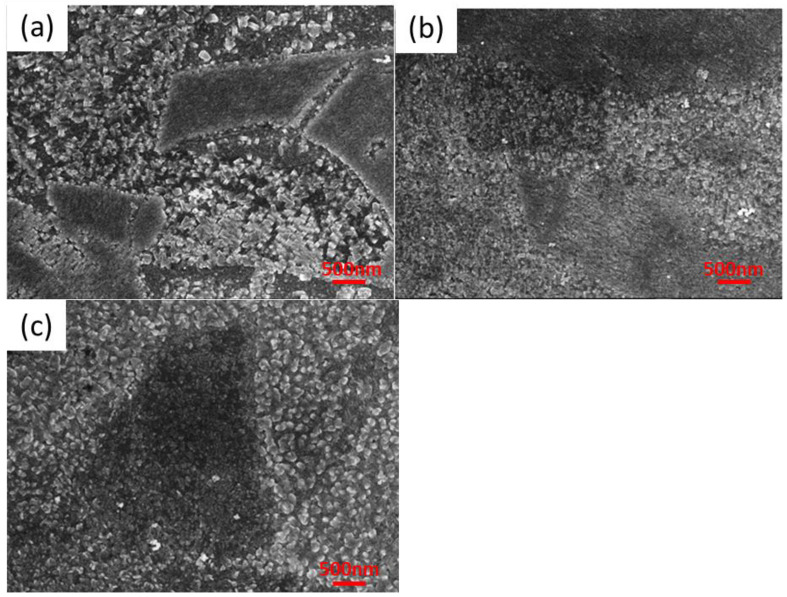
Surface morphology of the following samples: (**a**) SnS_PshL_104, (**b**) SnS_PshL_101, and (**c**) SnS_PshL_114 at magnification 5000×.

**Figure 4 materials-16-05019-f004:**
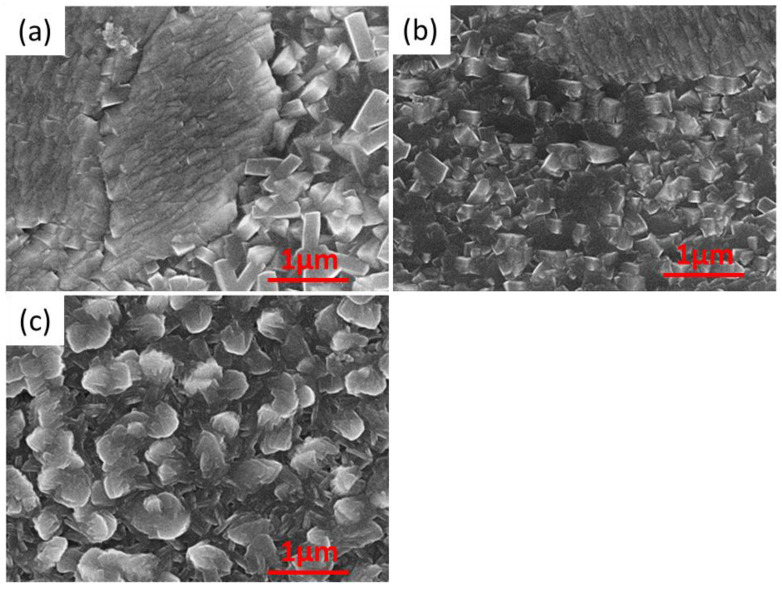
Surface morphology at 25000 magnification of the following samples: (**a**) SnS_PshL_104, (**b**) SnS_PshL_101, and (**c**) SnS_PshL_114.

**Figure 5 materials-16-05019-f005:**
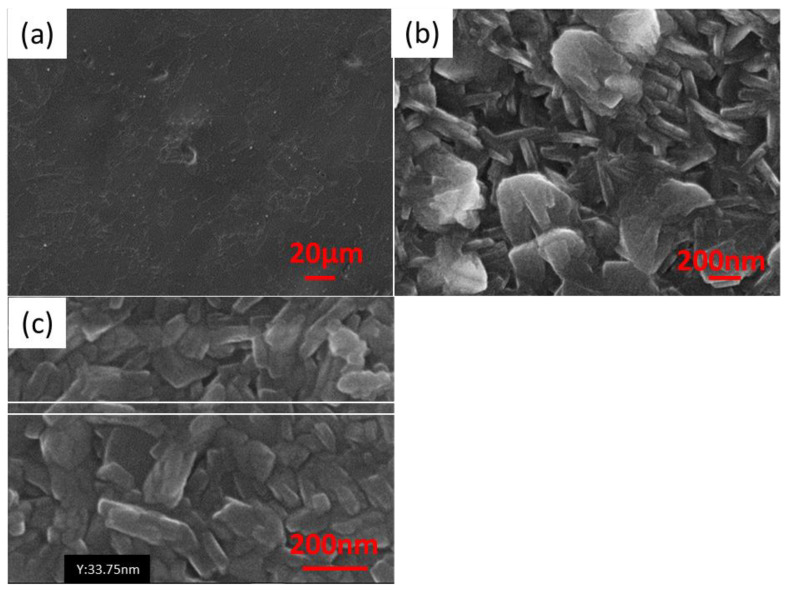
Surface morphology of SnS_PshL_113, at magnifications (**a**) 500, (**b**) 50,000, and (**c**) 100,000.

**Figure 6 materials-16-05019-f006:**
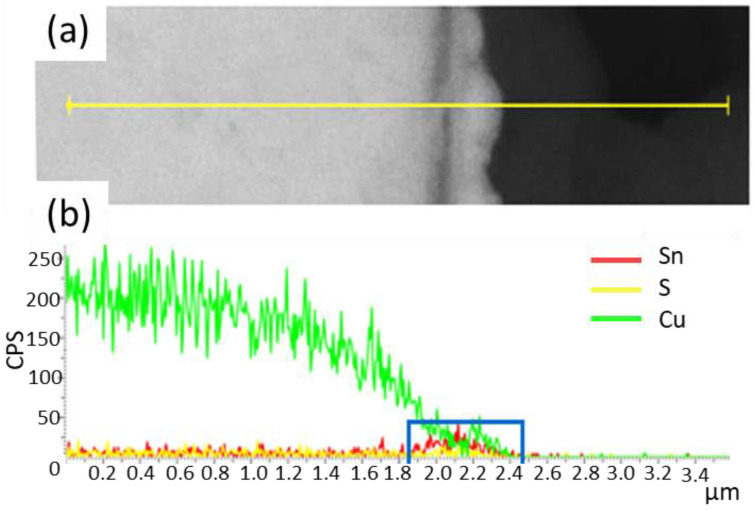
EDAX analysis SnS_Na_B011 sample, (**a**) SEM image showing the analyzed area, (**b**) results of analysis, where blue box highlights the SnS layer thickness.

**Figure 7 materials-16-05019-f007:**
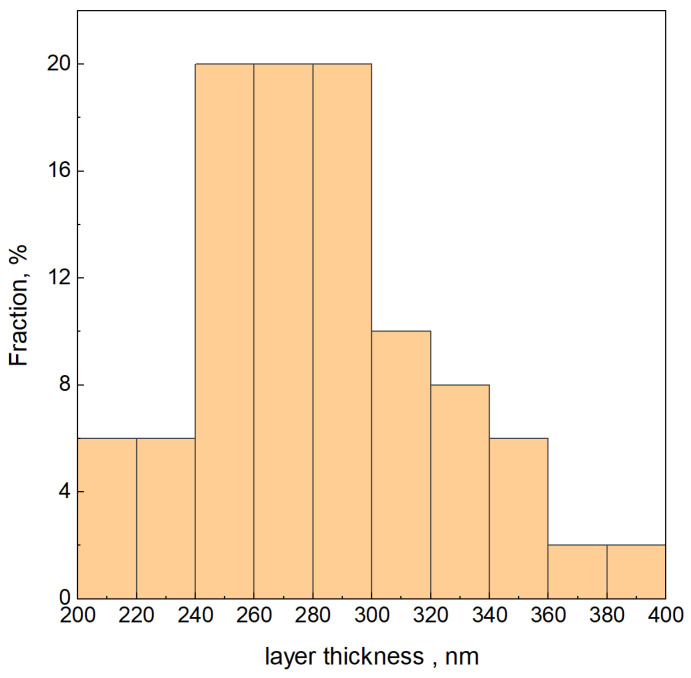
Testing the repeatability of the measurement for the selected sample SnS_Na_B010.

**Figure 8 materials-16-05019-f008:**
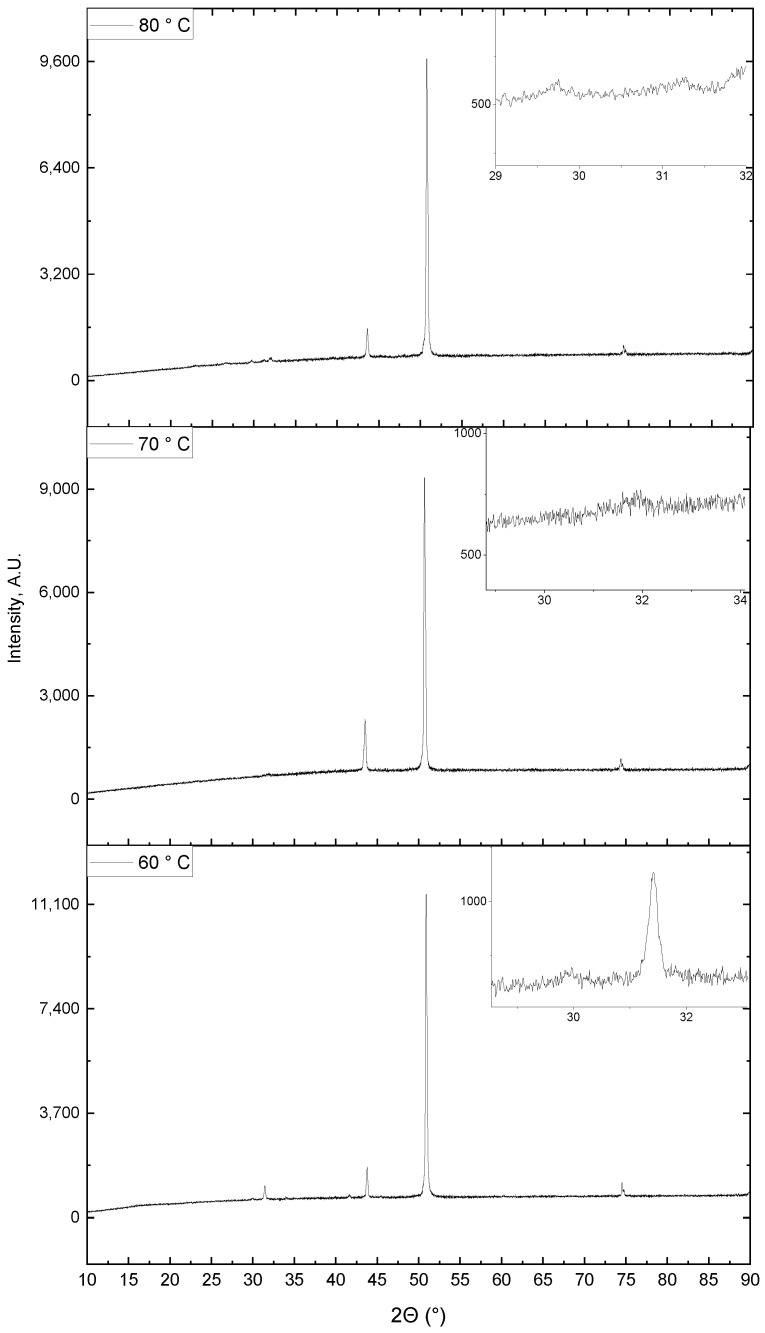
X-ray diffractograms of samples with a SnS layer on a copper substrate depending on the temperature of the deposition process.

**Figure 9 materials-16-05019-f009:**
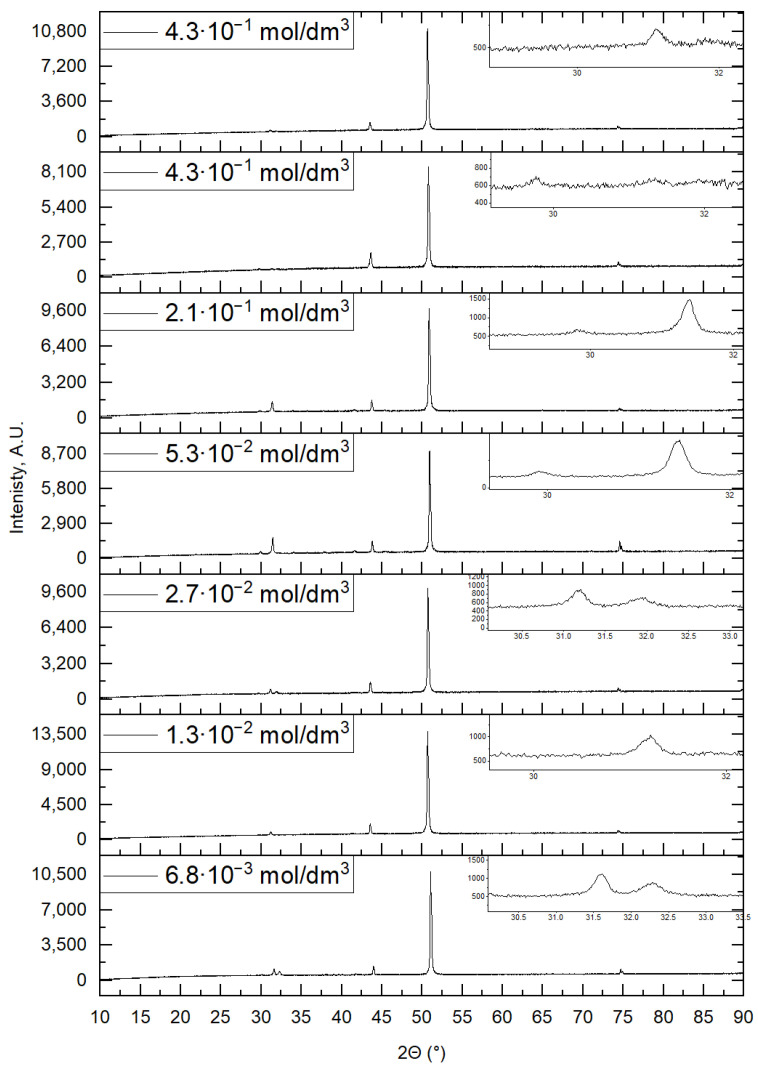
X-ray diffraction patterns of samples with an SnS layer on a copper substrate depending on changes in surfactant concentration.

**Figure 10 materials-16-05019-f010:**
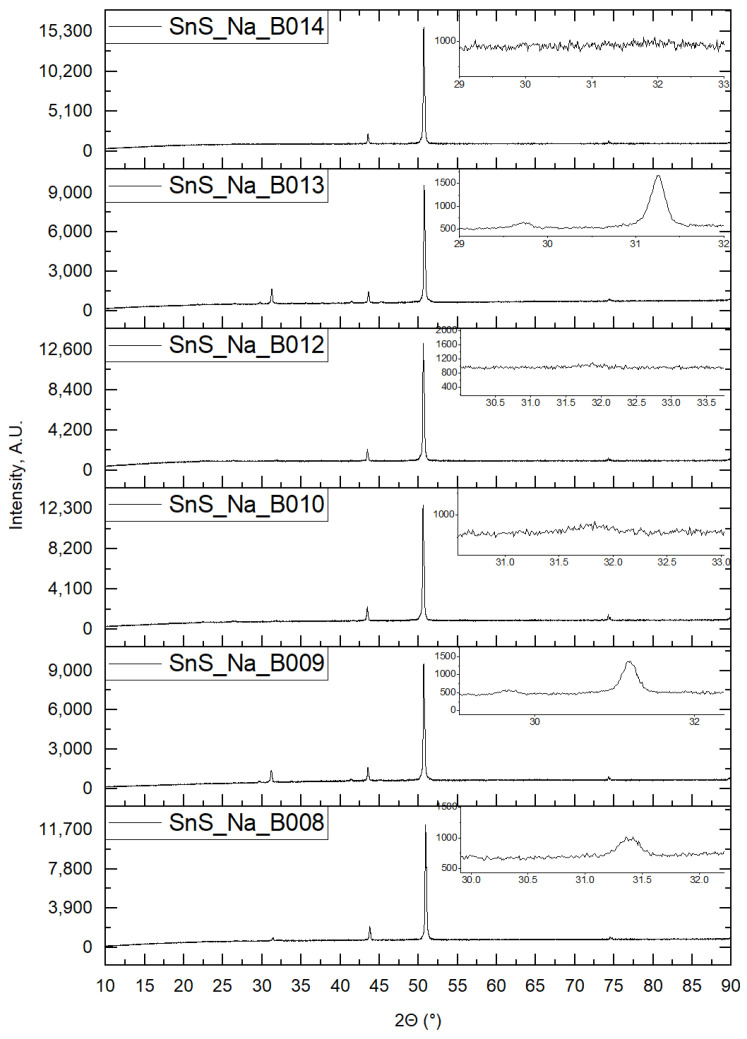
X-ray diffractograms of samples with an SnS layer on a copper substrate depending on the time of presence of sodium ions in the chemical bath during deposition.

**Figure 11 materials-16-05019-f011:**
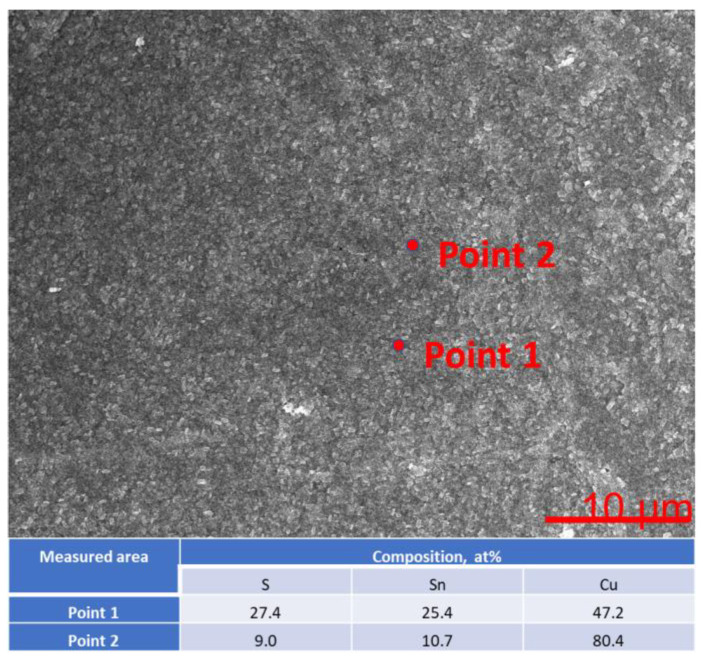
EDAX analysis of the sample.

**Figure 12 materials-16-05019-f012:**
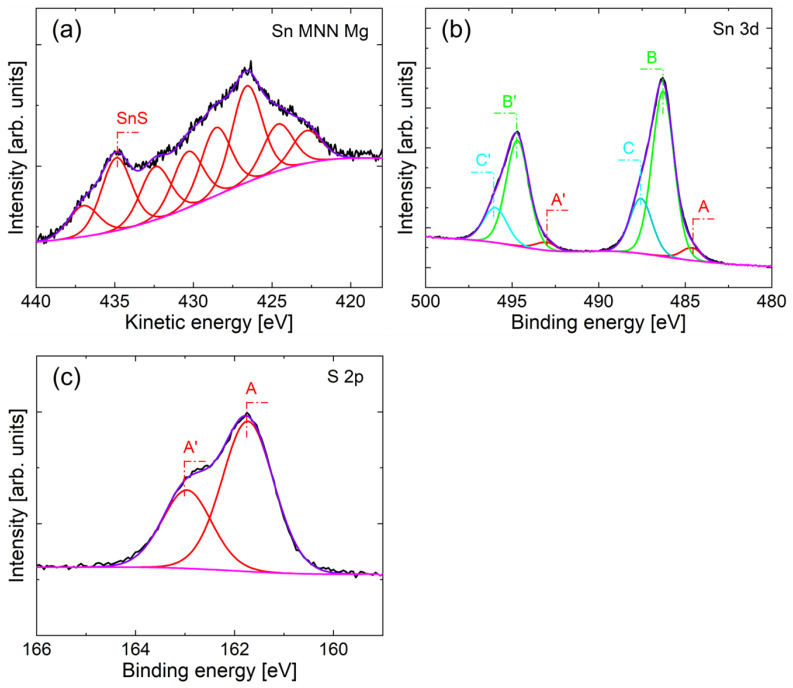
The deconvoluted XPS spectra of Sn M_4_N_45_N_45_ (**a**), Sn 3d (**b**), and S 2p (**c**) excitations of selected sample.

**Figure 13 materials-16-05019-f013:**
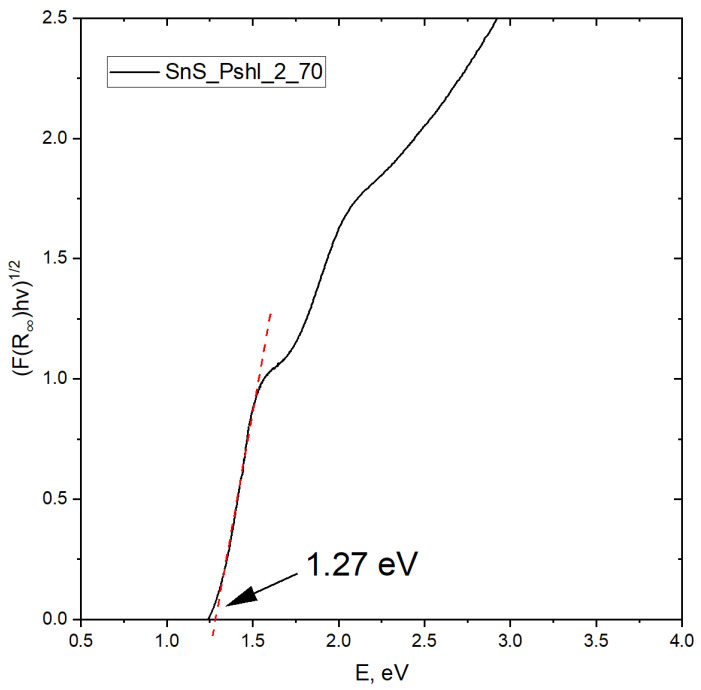
Method of band gap energy (Eg) determination from the Tauc plot.

**Table 1 materials-16-05019-t001:** A list of sample descriptions, including their specific synthesis parameters and visual observations of the finished samples.

Influence of Temperature		
Sample Label	Synthesis Temperature [°C]	Sample Size [cm^2^]
SnS_PshL_2_60	60	30 × 2
SnS_PshL_2_70	70	30 × 2
SnS_PshL_80	80	30 × 2
Influence of surfactant initial concentration		
Sample label	Surfactant concentration [g/dm^3^]	Sample size [cm^2^]
SnS_PshL_104	6.786·10^−3^	30 × 2
SnS_PshL_103	1.339·10^−2^	30 × 2
SnS_PshL_102	2.679·10^−2^	30 × 2
SnS_PshL_101	5.357·10^−2^	30 × 2
SnS_PshL_112	1.071·10^−1^	30 × 2
SnS_PshL_113	2.143·10^−1^	30 × 2
SnS_PshL_114	4.286·10^−1^	30 × 2
Influence of doping time		
Sample label	Doping time	Sample size [cm^2^]
SnS_Na_B008	5 min	30 × 5
SnS_Na_B009	3.5 min (synthesis time 3.5 min)	30 × 5
SnS_Na_B010	2 min	30 × 5
SnS_Na_B011	1 min	30 × 5
SnS_Na_B012	3 min	30 × 5
SnS_Na_B013	4 min	30 × 5
SnS_Na_B014	1 min 45 s (synthesis time 3.5 min)	30 × 5

**Table 2 materials-16-05019-t002:** Measurements of the thickness of the SnS layers on a copper substrate were obtained from the analysis of SEM images with the Image J^®^ program.

Sample Label	Layer Thickness [nm]	S.D. [nm]
SnS_PshL_80	295	95
SnS_PshL_2_70	383	35
SnS_PshL_2_60	-	-
SnS_PshL_114	327	48
SnS_PshL_113	319	43
SnS_PshL_104	160	24
SnS_PshL_103-2	202	40
SnS_PshL_102	272	42
SnS_PshL_101	276	44
SnS_Na_B008	342	51
SnS_Na_B009	279	38
SnS_Na_B010	283	41
SnS_Na_B011	213	40
SnS_Na_B012	237	28
SnS_Na_B013	265	32
SnS_Na_B014	-	-

**Table 3 materials-16-05019-t003:** Summary of measurements of the thickness of SnS layers in samples, made using an ellipsometer.

Sample Label	Layer Thickness [nm]	S.D. [nm]
SnS_PshL_80	223	7
SnS_PshL_2_70	228	3
SnS_PshL_2_60	235	2
SnS_PshL_114	238	1
SnS_PshL_114 (fiolet)	233	3
SnS_PshL_113	223	4
SnS_PshL_104	227	3
SnS_PshL_103-2	230	3
SnS_PshL_102	222	4
SnS_PshL_101	229	1
SnS_Na_B008	238	1
SnS_Na_B009	245	5
SnS_Na_B010	234	6
SnS_Na_B011	232	6
SnS_Na_B012	230	3
SnS_Na_B013	230	2
SnS_Na_B014	221	9

**Table 4 materials-16-05019-t004:** SnS thin layer resistance and calculated conductivity.

Influence of Temperature			
Sample Label	Synthesis Temperature [°C]	Resistance [Ω]	Conductivity [S/m]
SnS_PshL_2_70	70 °C	116.6	0.305
Influence of surfactant initial concentration			
Sample label	Surfactant initial concentration [g/dm^3^]	Resistance [Ω]	Conductivity [S/m]
SnS_PshL_114	4.286·10^−1^	212.0	0.164
SnS_PshL_101	5.357·10^−2^	164.0	0.215
SnS_PshL_102	2.679·10^−2^	178.0	0.205
SnS_PshL_103	1.339·10^−2^	0.2	146.997
SnS_PshL_104	6.786·10^−3^	53.8	0.663
Sodium doping			
Sample label	Doping time [min]	Resistance [Ω]	Conductivity [S/m]
SnS_Na_B008	5	198.0	0.172
SnS_Na_B011	1	26.0	1.342
SnS_Na_B012	3	96.6	0.365
SnS_Na_B013	4	168.0	0.210
SnS_Na_B014	1.75 (synthesis time 3.5 min)	53.4	0.279

**Table 5 materials-16-05019-t005:** Process conditions and characteristics of SnS thin films obtained from a chemical bath by the method of Chao Gao and others.

Grow Temperature [°C]	Na_2_S_2_O_3_/SnCl_2_ Ratio	S:Sn Ratio	Thickness [nm]	Conductivity [S/m]	D_hkl_ [nm]	E_g_ [eV]
60	1:1	46.7:53.3	350	5.57 × 10^−5^	25.5	–
60	2:1	46.8:53.2	375	1.01 × 10^−4^	26.7	1.13
60	3:1	48.6:51.4	380	1.52 × 10^−4^	27.3	1.26
80	1:1	47.5:52.5	325	1.78 × 10^−4^	28.1	–
80	2:1	48.7:51.3	435	1.82 × 10^−4^	27.2	1.01
80	3:1	48.9:51.1	450	1.88 × 10^−4^	28.9	1.09

## Data Availability

The datasets used and/or analyzed during the current study are available from Anna Komenda (annakomenda@student.agh.edu.pl) upon reasonable request.

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
