# Peer review of "Deposition of Thin Electroconductive Layers of Tin (II) Sulfide on the Copper Surface Using the Hydrometallurgical Method: Electrical and Optical Studies"

_materials, 2023, doi:10.3390/ma16145019_

Round 1

Reviewer 1 Report

Thin-film solar cells possess distinct advantages over crystalline silicon (Si) cells due to the variety and suitability of materials available for the construction of multi-junction systems that have enhanced efficiencies comparable to a Si modules. Thin films can be used on flexible substrates. Tin (II) sulfide is an attractive absorption material for low-cost thin-film solar cells due to its energy gap of 1.3 eV, high carrier mobility, high light absorption coefficient, and good p-type conductivity with solar conversion efficiency of 24%. Metal doping alters the energy gap and band positions in the SnS band structure and thus determines the electrical properties of the films due to metal substitution in the Sn vacancies. Herein, Komenda et al. investigated the deposit tin (II) sulfide thin films on a copper substrate using a chemical vapor deposition method. The effect of sodium (Na) doping in these films was studied. The authors found that the deposition parameters such as temperature, surfactant addition, and sodium doping time did not affect the thickness of the layers. The conductivity of SnS films was measured to be 0.3 S. The reported experimental result is fine for Materials after a major revision is made to remove many grammar errors and clarify the issues listed in follows:

(1) Lines 45-46, "15 for devices based on SnS mono junctions in their design15." What is the meaning of "15" in the sentence?

(2) In Abstract, the authors stated that the edges of the SnS grains were rounded off with the addition of a commercial surfactant. Please specify the commercial surfactant (give the name of the surfactant).

(3) The authors only characterized the synthesized thin film by the XRD, XPS and EDX methods. The manuscript lacks theoretical analysis of the experimental results.

(4) The photo-electrical properties of the synthesized Na-doped SnS thin film on Cu should be measured and reported for the application purpose.

(5) There are many grammar error and typos in the main text of the manuscript. <i> In Abstract, "The effect of sodium (Na) doping in these films studied" should be corrected to " The effect of sodium (Na) doping in these films was studied ". <ii> Lines 22-23, " The deposition parameters namely the effect of temperature, surfactant addition, and sodium doping time did not affect the thickness of the layers" should be corrected to " The deposition parameters such as temperature, surfactant addition, and sodium doping time did not affect the thickness of the layers ". <iii> Lines 95-96, What is the meaning of "However, reports which describe the influence of temperature, composition, and surfactants on the conductivity of the SnS layer in detail "? <iv> Line 363, "M. A. Dar, D. Govindarajan and G. N. Dar" should be corrected to "Dar, Govindarajan and Dar". <v> Line 348, "W. Y. Jim et al." should be corrected to "Jim et al.". <vi> Line 362, " almost 4 times thinner SnS layers " should be corrected to " almost 4 times thinner than SnS layers ". <vii> And please read the whole text carefully and remove all grammar errors.

(6) The format of the references in the Reference List does not match that of Materials. Please revise them according to Guide for Authors of Materials.

There are many grammar errors in the main text of the manuscript and please make an extensive revision for this issue.

Author Response

We are grateful to the reviewer for his thorough analysis of the manuscript and for catching numerous grammatical errors. We hope that our responses to the reviewer's comments are satisfactory. All changes in the manuscript are marked in red.

(1) Lines 45-46, "15 for devices based on SnS mono junctions in their design15." What is the meaning of "15" in the sentence?

Ad 1), Thank you for noting this error. The number “15” is error inserting text into publisher's template. The numbers were removed.

(2) In Abstract, the authors stated that the edges of the SnS grains were rounded off with the addition of a commercial surfactant. Please specify the commercial surfactant (give the name of the surfactant).

Ad 2) In the abstract, we omit this information, since it is detailed described in the part 2.1 of the manuscript line 103.  “A commercially available mixture of anionic surfactants, non-ionic surfactants, and amphoteric surfactants was used as the surfactant (Ludwik, Poland).”

(3) The authors only characterized the synthesized thin film by the XRD, XPS and EDX methods. The manuscript lacks theoretical analysis of the experimental results.

Ad 3) It is difficult to find theoretical models in this type of experimental work. After all, we are dealing with a reaction-diffusion system, and the problem is three-dimensional. Theoretical models can be made, but it takes hundreds of experiments to assemble them. At this stage, we conducted over 150 experiments (including repetitions). We still cannot show a meaningful convergent model.

Fair point, we are still working on this system and developing the potential of SnS PV.

(4) The photo-electrical properties of the synthesized Na-doped SnS thin film on Cu should be measured and reported for the application purpose.

Ad 4) Unfortunately, at this stage, we do not have the results of optical measurements. Based on conductivity measurements, which are directly related to the band gap, it can be concluded that this parameter is very important. We are planning a detailed study of the energy gap and reflection spectra of these layers, and further refinement of the method of dosing SnS with sodium.

(5) There are many grammar error and typos in the main text of the manuscript.

 <i> In Abstract, "The effect of sodium (Na) doping in these films studied" should be corrected to " The effect of sodium (Na) doping in these films was studied ".

<ii> Lines 22-23, " The deposition parameters namely the effect of temperature, surfactant addition, and sodium doping time did not affect the thickness of the layers" should be corrected to " The deposition parameters such as temperature, surfactant addition, and sodium doping time did not affect the thickness of the layers ".

<iii> Lines 95-96, What is the meaning of "However, reports which describe the influence of temperature, composition, and surfactants on the conductivity of the SnS layer in detail "?

<iv> Line 363, "M. A. Dar, D. Govindarajan and G. N. Dar" should be corrected to "Dar, Govindarajan and Dar".

<v> Line 348, "W. Y. Jim et al." should be corrected to "Jim et al.".

<vi> Line 362, " almost 4 times thinner SnS layers " should be corrected to " almost 4 times thinner than SnS layers ".

<vii> And please read the whole text carefully and remove all grammar errors.

Ad 5) We are grateful for pointing out a number of grammar errors. We commissioned a native speaker to proofread the language.

(6) The format of the references in the Reference List does not match that of Materials. Please revise them according to Guide for Authors of Materials.

Ad 6) the reference list was also corrected according to the Journal requirements.

Reviewer 2 Report

The manuscript "materials-2384846" by Komenda et al. reported Deposition of thin electroconductive layers of tin (II) sulfide on the copper surface using the hydrometallurgical method: electrical end optical studies. The authors have to make major changes. The authors should refer to the following comments to improve their work:

Abstract:
a. The term XPS  is abbreviated in the abstract without explanation, please revise and rewrite.

Introduction:
a. Page 2, line 46: 15 should be superscript.

b. Please, review the results of several other studies in the introduction and explain the difference between this study and the from others.

Results:

a) (3.1. Influence of Deposition temperature)
1. Why does increasing temperature reduce the number of surface defects? (Explain in the manuscript, don't just report)

b) (3.2. Influence of surfactant addition)
2. In your opinion, what is the reason for the rounding of crystals with increasing surfactant concentration? It seems that in addition to being rounded, they also became wider (Figure 4b). It seems that if the concentration of surfactant is increased, the circles (crystals) become closer together (integrate), although this is speculation. It seems that increasing the surfactant concentration changes the crystal growth direction.

c) The title of sections 3.2 and 3.3 is the same.

d) In Figure 5, the authors reported a thickness of about 75 nm. In fact, this thickness changes in different places. It is better to report a minimum and maximum value.

e) The placement of Figures 1 to 5 is incorrect. First, write the entire paragraph related to each Figure and then place the Figure.

f) The amount of crystallinity should be reported.

g) The quality of the Figures is very low.

h) Improve the discussion. In many sections, only the results report can be seen.

i) The language of the manuscript should be checked. In many sections, the sentences should be integrated. In some sections, the sentences are very short.

Moderate editing of English language.

Author Response

We are grateful to the reviewer for his thorough analysis of the manuscript and for catching numerous grammatical errors. We hope that our responses to the reviewer's comments are satisfactory. All changes in the manuscript are marked in red.

Abstract:
a. The term XPS  is abbreviated in the abstract without explanation, please revise and rewrite.
Ad a) Thank you for noting this error. The abbreviation was explained.
Introduction:
a. Page 2, line 46: 15 should be superscript.
Ad a) Thank you for noting this error. The number “15” is an error in inserting text into the publisher's template. The numbers were removed.
b. Please, review the results of several other studies in the introduction and explain the difference between this study and the from others.
ad b) Such a statement was made and included in the results and discussion section. Usually, the results from the current research are not shown in the introduction section.
Results:

a) (3.1. Influence of Deposition temperature)
1. Why does increasing temperature reduce the number of surface defects? (Explain in the manuscript, don't just report)
Ad 1) we add a section describing the possible mechanisms of temperature influence.

“The effect of temperature on reducing the number of defects should be associated with a change in solution viscosity. The process is carried out in a heterogeneous system. So the rate-limiting element may be transported at the interface. In such a situation, if the slowest step is diffusion, the viscosity can be changed by adding e.g. alcohol, increasing the temperature, or increasing the intensity of mixing. However, we would like to point out that the detailed study of the mechanism is not the subject of this work, but the preliminary selection of the conditions for obtaining such layers.”

  1. b) (3.2. Influence of surfactant addition)
    In your opinion, what is the reason for the rounding of crystals with increasing surfactant concentration? It seems that in addition to being rounded, they also became wider (Figure 4b). It seems that if the concentration of surfactant is increased, the circles (crystals) become closer together (integrate), although this is speculation. It seems that increasing the surfactant concentration changes the crystal growth direction.

Ad 2)  It seems that if the concentration of surfactant is increased, the circles (crystals) become closer together (integrate). Increasing the surfactant concentration may change the crystal growth direction. Such an effect is frequently observed and reported in the literature. The best example is gold nanorods formation in the presence of CTAB (Cetyltrime-thylammonium bromide). – this part was added to the section 3.2.

c) The title of sections 3.2 and 3.3 is the same.
Ad c) thank you for noting this error. We corrected this part.
d) In Figure 5, the authors reported a thickness of about 75 nm. In fact, this thickness changes in different places. It is better to report a minimum and maximum value.
Ad d) we correct this part. Adding the range +/-20 nm
e) The placement of Figures 1 to 5 is incorrect. First, write the entire paragraph related to each Figure and then place the Figure.
Ad e) We are grateful for this suggestion. Now the manuscript is more clear
f) The amount of crystallinity should be reported.
Ad f) Sorry, but I don't understand the question. Does the reviewer mean a degree of crystallinity? How can the degree of crystallinity affect the cell based on SnS? If there is no such correlation, does it make sense to conduct such research? We will be grateful for suggestions in this regard. We do not usually conduct such research and we have no experience in this field.
g) The quality of the Figures is very low.
Ad g) In all SEM images the brightness and contrast have been corrected to make the photo more clear.
h) Improve the discussion. In many sections, only the results report can be seen.
Ad h) The discussion part was extended taking into account obtained results as well as data from the literature.
i) The language of the manuscript should be checked. In many sections, the sentences should be integrated. In some sections, the sentences are very short.

Ad i) We are grateful for pointing out a number of grammar errors. We commissioned a native speaker to proofread the language.

Reviewer 3 Report

The manuscript ‘Deposition of thin electroconductive layers of tin (II) sulfide on the copper surface using the hydrometallurgical method: electrical end optical studies’ authored by Komenda et al. deposited SnS thin films on the copper substrates via a chemical bath method by tuning different parameters such as temperature, surfactant addition and Na doping. The work seems to be interesting, however, there are several big scientific flaws in its current form. Thus, I think this manuscript is unpublishable unless the authors make a major revision for further consideration.

A couple of comments:

1.     The paper structure seems to be quite messy and lacks deep discussion. It rather looks like a report not a scientific paper. The authors should make significant efforts to rearrange the paper structure with sufficient discussion along with the results.

2.     The Introduction section is written in an unclear order. The knowledge gap and novelty of this work should be clearly described after the relevant literature review.

3.     No description for materials used in this work as well as for the characterization methods.

4.     The quality of the figures is low. The authors should make sure all figures are visible with sufficient information either in legend or in captions.

5.     There are lots of typos and grammar errors, which require the authors to thoroughly check and rectify. A professional language checking is highly recommended.

6.     More characterizations are required such as optical property and electrical property.

Extensive editing of English language required.

Author Response

We are grateful to the reviewer for his thorough analysis of the manuscript and for catching numerous grammatical errors. We hope that our responses to the reviewer's comments are satisfactory. All changes in the manuscript are marked in red.

  1. The paper structure seems to be quite messy and lacks deep discussion. It rather looks like a report not a scientific paper. The authors should make significant efforts to rearrange the paper structure with sufficient discussion along with the results.

Ad 1) additional discussion related to the obtained data was added to the manuscript.

  1. The Introduction section is written in an unclear order. The knowledge gap and novelty of this work should be clearly described after the relevant literature review.

Ad 2) We found the introduction chaotic. First of all, we did not specify the purpose of the publication. We fixed our mistake.

  1. No description for materials used in this work as well as for the characterization methods.

Ad 3) We are grateful for pointing out this lack of information. We add this description to the Materials and Methods section.

  1. The quality of the figures is low. The authors should make sure all figures are visible with sufficient information either in legend or in captions.

Ad 4) In all SEM images the brightness and contrast have been corrected to make the photo more clear.

  1. There are lots of typos and grammar errors, which require the authors to thoroughly check and rectify. A professional language checking is highly recommended.

Ad 5) We are grateful for pointing out a number of grammar errors. We commissioned a native speaker to proofread the language.

  1. More characterizations are required such as optical property and electrical property.

Ad 6) We have provided conductivity data, these are basic electrical parameters. The Hall effect and other electrical effects are related to conductivity. We agree, in fact, there is no data on optical properties in our work. This is due to the fact that it would excessively expand the publication and make it difficult for readers to keep up with the text. You cannot optimize two parameters at the same time. Obtaining a sufficiently thin layer with the right composition is required for the SnS layer to be used for PV. In the next stages, optical properties can be optimized by adjusting the concentration of impurities and the degree of crystallization (annealing), etc.

Reviewer 4 Report

In this paper, the authors used a chemical bath method to deposit thin films of tin (II) sulfide onto a copper substrate and studied the effect of sodium doping. They used various characterizations to confirm their methodology. The article needs some major improvements, such as;

1.       In abstract art, please write the motivation for this work.

2.       In the introduction part, please add comprehensive details related to this work published recently, and clearly explain your idea in the last paragraph.

3.       The introduction part is presently in unorganized form; please organize it briefly.

4.       Please add some recent literature relevant to this study in support of your methodology.

5.       Please provide references in the methodology part about your preparation methods or techniques if you think it's not new.

6.       Figure 1-5, please add a bar scale.

7.       I think you can merge figures 1-5 into one figure as a,b,c,d,e part to make better comparison and to provide ease for the readers to understand this work quickly.

8.       The figure, better resolution needed. It's hard to read the axis.

9.       Figures 8 and 9, 2-Theta symbol needs to be correct.

10.   Figure 12, poor resolution. The figure needs to change.

11.   Please add key findings in the conclusion part.  

Extensive editing of English language required

Author Response

We are grateful to the reviewer for his thorough analysis of the manuscript and for catching numerous grammatical errors. We hope that our responses to the reviewer's comments are satisfactory. All changes in the manuscript are marked in red.

  1. In abstract art, please write the motivation for this work.

Ad 1) In our opinion, the abstract is not the place to describe the motivation to undertake research. We agree with the reviewer, our mistake is the lack of a clear statement and definition of the purpose of the work. We added this part to the introduction section.

  1. In the introduction part, please add comprehensive details related to this work published recently, and clearly explain your idea in the last paragraph.

Ad 2. We agree with the reviewer, our mistake is the lack of a clear statement and definition of the purpose of the work. We added this part to the introduction section.

  1. The introduction part is presently in unorganized form; please organize it briefly.

Ad 3) We have made changes to the introduction, we hope that now the construction is clearer and clearer.

  1. Please add some recent literature relevant to this study in support of your methodology.

Ad 4) We tried to present the widest and latest knowledge in the field of SnS synthesis. However, we have not encountered any work on the synthesis of SnS thin films on a Cu substrate. We will be grateful for any suggestions on what works are worth quoting on this topic.

  1. Please provide references in the methodology part about your preparation methods or techniques if you think it's not new.

Ad 5) To our knowledge, conductivity measurements of SnS layers have been performed for the first time in this way. So far, SnS has been deposited on non-conductive or weakly conductive substrates. Then using the typical four-blade method is simple. In the case of SnS deposited on Cu, such a measurement cannot be performed. Therefore, we proposed to use copper tape on the SnS side and measure the resistance, as in a typical system of interconnected resistors. Thanks to the reviewer's note, we noticed that we didn't cover this part in the experimental section. Therefore, this part has been significantly expanded.

  1. Figure 1-5, please add a bar scale.

Ad 6) The scale bar is included in all the mentioned drawings. The software does this automatically.

  1. I think you can merge figures 1-5 into one figure as a,b,c,d,e part to make better comparison and to provide ease for the readers to understand this work quickly.

Ad 7. We deliberately separated these images so that readers can easily follow the influence of individual parameters on the morphology of the samples. In addition, we do not have all photos at the same magnification. Comparing such photos would lead to complete confusion.

  1. The figure, better resolution needed. It's hard to read the axis.

  1. Figures 8 and 9, 2-Theta symbol needs to be correct.

  1. Figure 12, poor resolution. The figure needs to change.

Ad 8, 9, and 10) Some drawings were indeed of low quality. Especially XPS and EDS. we have corrected these drawings. We hope that now they are more aesthetic and legible.

  1. Please add key findings in the conclusion part.

Ad 11) The last chapter, i.e. the conclusions, has been slightly changed to emphasize the results of our work.

Round 2

Reviewer 1 Report

The authors have revised their manuscript carefully by considering all the comments and suggestions from the Reviewers. Now the revised manuscript is adequately good for publication. Therefore, I would like to recommend it for publication.

English is fine.

Author Response

Thank you for your cooperation and fruitful comments. We made further adjustments to the English language. All changes are marked in red in the manuscript.

Reviewer 2 Report

Accept in present form.

Minor editing of English language required.

Author Response

Thank you for your suggestions and fruitful comments. We made further adjustments to the English language. All changes are marked in red in the manuscript.

Reviewer 3 Report

Although the author made some effort to address some points, the quality of the manuscript is still low. The revision looks superficial which has not intrinsically optimized the manuscript. Even the typo of 'end' (should be 'and') is still in the title. Many figures are still not very visible. The structure of the manuscript is still not clear. The authors tried to explain why they did not investigate the optical property, however it is weird to state 'optical studies' in the title but eventually without showing any relevant data in the manuscript. Based on above concerns, I cannot recommend accepting this manuscript in its current form.

Extensive editing of English language required as there are still lots of typos and grammar errors.

Author Response

Thank you for your suggestions and fruitful comments. We made further adjustments to the English language. All changes are marked in red in the manuscript.
As you can see, the amount of changes is significant. We hope that now the publication meets the high requirements of the reviewer.
Considering the original title, we included a chapter in the work in which we present the calculation of the band gap for an example sample.

We are unable to address only the reviewer's comment regarding "Many figures are still not very visible."
Does the reviewer mean the sharpness of the image or the size of the images? We would like to point out that 3 out of 4 reviewers accepted the current quality of the photos. We can correct them, but we must receive clear information about what exactly the changes are about. The previous drawings were indeed of poor quality. We hope that the publication in its current form will be accepted by the Reviewer.

Reviewer 4 Report

Thanks for the revision. I am supportive of its publication in its current form. 

Moderate editing of English language required

Author Response

(The authors gave the same response as above.)
